# Field-Flow Fractionation in Molecular Biology and Biotechnology

**DOI:** 10.3390/molecules28176201

**Published:** 2023-08-23

**Authors:** Stefano Giordani, Valentina Marassi, Anna Placci, Andrea Zattoni, Barbara Roda, Pierluigi Reschiglian

**Affiliations:** 1Department of Chemistry “Giacomo Ciamician”, University of Bologna, 40126 Bologna, Italyvalentina.marassi@unibo.it (V.M.);; 2byFlow srl, 40129 Bologna, Italy

**Keywords:** asymmetrical flow field-flow fractionation (AF4), laser scattering, native separation, cell sorting, pharmaceutics, bio-nanoparticles, molecular biology, separation science

## Abstract

Field-flow fractionation (FFF) is a family of single-phase separative techniques exploited to gently separate and characterize nano- and microsystems in suspension. These techniques cover an extremely wide dynamic range and are able to separate analytes in an interval between a few nm to 100 µm size-wise (over 15 orders of magnitude mass-wise). They are flexible in terms of mobile phase and can separate the analytes in native conditions, preserving their original structures/properties as much as possible. Molecular biology is the branch of biology that studies the molecular basis of biological activity, while biotechnology deals with the technological applications of biology. The areas where biotechnologies are required include industrial, agri-food, environmental, and pharmaceutical. Many species of biological interest belong to the operational range of FFF techniques, and their application to the analysis of such samples has steadily grown in the last 30 years. This work aims to summarize the main features, milestones, and results provided by the application of FFF in the field of molecular biology and biotechnology, with a focus on the years from 2000 to 2022. After a theoretical background overview of FFF and its methodologies, the results are reported based on the nature of the samples analyzed.

## 1. Introduction

### 1.1. FFF Techniques

Field-flow fractionation (FFF) techniques are separative and analytical platforms (if coupled to a suitable detector) first introduced by J. C. Giddings in the 1960s [1]. They have applications in the separation and study of dispersions in a wide range of sizes from small peptides [2] to particles above the micron (e.g., cells) [3].

FFF instrumentation is essentially configured as a traditional HPLC system: pumps for generating the eluent flow followed by the sample injector, the separative element (the channel), and finally the detectors. Figure 1 shows a typical instrumental configuration and the types of fields that have been tested or conceptualized for use in FFF. The fields in bold gave rise to the subtechniques that are currently commercially available (FlFFF, SdFFF, and ThFFF) or have been significantly employed. Instrumentation for different subtechniques requires the use of peculiar components and solutions related to the nature of the field, making each instrument configuration specific to the application of a particular subtechnique [1]. Further details on the different subtechniques are provided in the following section and the references therein. The peculiarity of FFF systems consists of the separative element, a capillary channel often with a rectangular section. The fractionation mechanism is single-phase-based, as the channel does not contain a stationary phase [4]. The separation is achieved by the combined action of the laminar flow of eluent (with parabolic velocity profile) coaxial to the channel axis and an external field applied perpendicularly and directed toward a channel wall called the “accumulation wall”. Sample components (the analytes) differing in molar mass, size, and/or other physical properties are driven by the applied field into different velocity regions within the parabolic flow profile of the mobile phase across the channel. Parabolic flow is characterized by maximum velocities in the center of the channel decreasing until reaching zero at the walls [5].

Based on how the analytes are distributed across the channel, different operating modes [6,7] are distinguished (Figure 2):

Normal mode is the result of the balancing process: a drift velocity applied to the analytes by the external field toward the accumulation wall and their opposing diffusivity. Smaller analytes are distributed nearer to the center of the channel compared to the bigger ones. Consequently, the smaller the analyte, the lower its retention time. This operating mode affects the elution of macromolecules and nanoparticles.

Reverse mode is typical for micron-sized nanoparticles whose diffusion from the accumulation wall is negligible. In the ideal steric mode, regardless of their size, the analytes are driven by the external force to touch the accumulation wall. Consequently, bigger analytes are affected overall by higher flow vectors compared to the smaller ones generating an opposite elution order compared to normal mode. In real settings, such as particularly at high elution flow rates, hyperlayer mode takes place: particles are affected by lift forces (caused by the fluxes in the channel), driving them away from the wall at a distance higher than their diameter. The elution order is still reversed compared to normal mode, but retention depends not solely on the size of the analytes (like in steric mode) but also other physical features (e.g., shape, rigidity, density, surface properties).

FFF separation range (about 15 orders of magnitude mass-wise, corresponding to a few nanometers to about 100 µm in size) derives from the possibility of changing the nature of the force field or gradient applied. Various FFF variants are thus distinguished based on different chemical–physical properties of the samples exploited for the separation.

Thermal FFF (ThFFF). The field is generated by a temperature gradient created by two blocks with high thermal conductivity, which constitute the walls of the channel. The separation is performed according to the analyte’s Soret coefficient or the ratio between the thermal (D_T_) diffusion coefficient and the hydrodynamic diffusion coefficient (D). ThFFF has been mainly used in polymer separation [8].

Flow FFF (FlFFF). A second flow, applied perpendicularly to the main flow by crossing the channel transversely, drives the analytes toward the accumulation wall according to their diffusivity (correlated to their hydrodynamic radius). This flow is usually called crossflow. FlFFF represents the most successful and exploited FFF variant due to its ability to provide itself uncorrelated mass-hydrodynamic size information [9] and its wide operational range in the nanoscale world. It is possible to distinguish three main variants of FlFFF:

Symmetrical FlFFF (SF4). The transverse flow crosses the channel through both porous channel walls and it is generated by a second pumping system regardless of the elution flow. Nowadays, SF4 has been replaced by other FlFFFs (discussed below) due to providing higher resolutions and faster separation speeds over SF4 [10].

Asymmetrical FlFFF (AF4). The channel has only one porous wall corresponding to the accumulation one. The flow entering the channel is split into two components: a longitudinal flow, which drags the sample along the channel, and a transverse flow, which crosses the accumulation wall and generates the fractionation field [11]. The typical channel is characterized by a trapezoidal shape and uniform thickness, which are defined by placing specific spacers between the accumulation wall and the upper wall of the channel. Other variants exploiting different channels have been developed such as the well-established hollow-fiber channel (discussed below), the frit-inlet channel [12], and, most recently, a thickness-tapered channel [13]. In the frit-inlet channel, the sample is relaxed hydrodynamically as it enters the channel, omitting the focusing step which, in some cases, could cause sample aggregation/loss. The latter, although still in the prototype status, seems to offer certain advantages in steric/hyperlayer mode because it induces an increase in hydrodynamic lift forces, which can be useful for eluting long-retained particles without the need for field programming or reduced channel thickness.

Hollow-Fiber FlFFF (HF5). A miniaturized variant of AF4, in which the channel is replaced by a porous tubular fiber with a circular section [14]. Compared to AF4, this variant typically has greater (or in any case comparable) efficiency and greater detectability (there is less dilution of the sample during separation) [15]. Moreover, the separative channel is cheap, and it is possible to work in disposable mode, avoiding the risk of cross-contamination. These features are very important if a downstream analysis [16] of the collected fractions is necessary or in pharmaceutical quality control [14] However, HF5 is not suitable for the injection of large quantities of samples, which reduces its applications on a preparative scale. An improvement in throughput is represented by MxHF5, which consists of dozens of HF5 parallel modules [17].

Electrical FFF (ElFFF). An electric field is created by two blocks with high electrical conductivity (typically graphite) appropriately charged, which also constitutes the walls of the channel [18]. The separation is based on their electrophoretic mobility, which is correlated to their charge. Pure ElFFF has been abandoned due to poor resolution and alteration of the signal caused by electrolysis products and bubbles generated during the experiments [19]. Recently a new variant, electrical asymmetrical flow field-flow fractionation (EAF4), has been developed [20,21]. EAF4 combines both ElFFF and AF4, enabling separation based on both diffusion coefficient (based on AF4) and, to some extent, the surface charge of analytes (based on ElFFF). Ideally, EAF4 would provide charge-size dependent separation of samples with different charges or charge densities, even with the same size, potentially improving the resolution between components compared to conventional AF4 due to the utilization of the electric field. A potential disadvantage of EAF4 is the increased number of parameters affecting the result of the separation due to the combination of the two fields applied.

Sedimentation FFF (SdFFF) or Centrifugal FFF. The channel is folded circularly and rotated around its axis. In this way, a centrifugal force is created radially with respect to the rotation axis and perpendicular to the flow of the mobile phase. The separation is based on the inertial mass, the diameter of the analytes, and the difference in density between the analyte and the carrier [22].

Gravitational FFF (GrFFF). It is a very simple and inexpensive technique that uses only Earth’s gravity as an applied field. It is mainly used to separate large analytes (around 1–100 μm) raging from inorganic particulate matter to particles of biological interest (such as cells and bacteria) [23]. A sub-variant is represented by the use of a split-flow thin cell (SPLITT), which is able to provide a quick continuous binary separation on a preparative scale [24].

Magnetic FFF (MgFFF). This technique is exploited for the separation and characterization of magnetic nanoparticles alone or even when their surface is coupled with other systems (ex., antibodies for specific binding to biological cells or chemotherapeutic drugs for magnetic drug delivery). MgFFF exploits a magnetic field to obtain particle separation based on the magnetic dipole moments of the nanoparticles. However, during the process, the analytes can interact with each other and perturb the separation. Different kinds of separative channels and magnetic field generation have been developed based on this approach [25,26].

Dielectrophoresis FFF (DIFFF or DEPFFF). These techniques exploit an electric field to separate polarizable objects. The field induces a distribution of the superficial charges of the analytes, generating inducted dipoles. Due to the electric field applied being non-uniform, the analytes (the inducted dipoles) experience a net force, causing their translation to different portions of the channel from the accumulation wall. This force is called the dielectrophoretic force and it is related to the dipole moment of the analyte. This technique has been used to separate a wide array of species, from the fractionation of colloidal nanoparticles to the isolation of diseased cells from healthy ones [27].

The main features of the FFF subtechniques available are listed in Table 1.

Other FFF variants based on other fields have been theorized. Of them, only acoustical FFF has been developed and tested [28]. Although it has not been imposed as a standalone FFF variant, it has recently been exploited to improve the resolution of GrFFF [29].

**Table 1 molecules-28-06201-t001:** Main features of the most common FFF techniques.

Technique	Separation Based on Analytes:	Typical Size Range	Main Applications	
FlFFF	Hydrodynamic diffusion coefficient	1 nm–20 μm	Proteins, polysaccharides, lipids, nanoparticles, micelles, vesicles, organelles, polymers	[30,31]
ElFFF	Electrophoretic mobility, size	1 nm–1 μm	Nanoparticles, proteins, viruses, bacteria	[19,32]
ThFFF	Thermal diffusivity, size	5 nm–10 μm	Polymers, gels, nanoparticles	[33]
GrFFF	Size, density	1–100 μm	Cells, bacteria, organelles	[5]
SdFFF	Size, density	10 nm–50 μm	Cells, bacteria, organelles	[5]
MgFFF	Magnetic properties	10–20 μm	Magnetic nanoparticles	[34]
DlFFF	Size, dielectric properties	1–20 μm	Cells	[27]

### 1.2. FFF and SEC Complementarity

Size exclusion chromatography is the most common separation technique for size-based separation of biological nanosystems [35,36]. A common leitmotiv in FFF studies concerns the substitution of SEC multidetection with FFF multidetection (especially AF4) due to the limitations of the first technique and the advantages reported by the second. SEC is characterized by a limited size resolution range (max. 10^7^ Da, ~100 nm). The presence of a stationary phase can also induce aggregation of the sample and impose limitations on mobile phase composition, which must be compatible with the column. Moreover, it is possible to observe adsorption and/or dissociation processes caused by the interaction of the sample with the stationary phase [37]. On the other hand, FFF techniques are characterized by a higher dynamic interval, and the absence of a mobile phase prevents the problems of SEC, allowing it to work in native conditions [11,16].

### 1.3. Detectors (Online Coupling)

The real potential of an FFF platform, besides the mere separation power, is expressed by the downstream coupling with an array of detectors providing multiparametric information on the separated species. Below, the most common detectors coupled online with an FFF platform are briefly described, except for the common UV/Vis spectrophotometer.

#### 1.3.1. Multiangle Light Scattering (MALS) and Dynamic Light Scattering (DLS)

MALS is a characterization technique that determines the molar mass (MW), gyration radius (Rg), and particle size distribution (PSD) of the analytes by collecting light scattered by particles in suspension at different angles. If sample concentration and the specific refractive index increment (dn/dc) are measured or known, MALS can measure MW without reference to calibration [38]. By plotting log Rg over log Mw, it is possible to obtain the conformation plot, whose slope (called *v*-value) gives information on the shape of the nanosystems [39,40]. Alternatively, the molecular shape can be estimated by the ratio between Rg and Rh [41]. By comparing these parameters, it is also possible to calculate the apparent densities for the single components, which may also be helpful for providing a detailed characterization [42]. The required Rh values can be obtained through FlFFF theory using the Stokes–Einstein equation. For other FFF techniques, DLS coupling is required. This detector evaluates the exponential decay in the scattered light intensity by the construction of an Rh-dependent autocorrelation function based on the Stokes–Einstein equation. From the analysis of such function, it is thus possible to evaluate the hydrodynamic radius/diameter of the nanoparticles. When the nanoparticles are not spherical, of uniform density, and hard, as considered by the Stokes–Einstein equation, the Rh calculated with both techniques has to be considered equivalent Rh. [43]. The Rg/Rh ratio values and resulting general conformation are detailed, together with the corresponding v value, in Table 2.

#### 1.3.2. Differential Refractive Index (dRI) Detector

The dRI detector is based on the method of continuously measuring the difference in refractive index (n) between the sample flowing through a flow cell and the eluant located in the reference cell. Provided the knowledge of the dn/dc parameter for the analyte in the experimental environment (obtainable through calibration), it is possible to associate the dRI signal with the sample’s concentration. In a typical FFF multidetector platform, this detector is used with or without a UV/VIS detector as a “concentration” source for MALS calculations. Compared to a UV/VIS detector, the dRI is more difficult to handle since it is characterized by lower sensitivity and subjected to higher signal fluctuations caused by flow changes. However, dn/dc changes little with wavelength and sample type compared to the extinction coefficient. dRI is typically preferred for substances (such as polysaccharides) that only slightly absorb the UV/VIS light, whose response may also be greatly influenced by scattering contributions [44].

#### 1.3.3. Fluorescence Detector (FLD)

A Fluorescence Detector (FLD) is also a commonly used detector, The high selectivity of the excitation/emission process at proper wavelengths makes the FLD specific for certain organic substances. Moreover, the detector is characterized by high sensitivity (tunable by regulating the excitation intensity) and high stability against possible drifts caused by pressure/flow changes. Online coupling of the FLD and AF4 can be used to detect the fluorescent components in the samples, ranging from proteins to fluorescent dyes for targeted recognition [45].

#### 1.3.4. MS Detectors

Inductively coupled plasma mass spectrometry (ICP-MS) is the most common MS detector coupled online to an FFF platform. This technique uses a high-temperature ionization source with a sensitive and rapid scanning mass spectrometer. It is characterized by higher detectability compared to classical detectors [46]. For example, the incorporation of ICP-MS for the analysis of metal-containing engineered NPs provides lowering the particle concentration detection limits down to the μg/L range, which is far below those provided by commonly used detectors [47]. It is extensively used for the detection of low-concentration nanomaterials (mostly metal NPs), providing an additional dimension (elemental composition) to the FFF platform [48] (Figure 3). FFF-ICPMS can also provide information about natural colloids in aquatic systems, their ability to bind and transport trace elements, and size-dependent variations in their composition [49] (Figure 3).

ESI/TOF-MS has also been successfully coupled online with FlFFF (HF5 and AF4), allowing us to observe native or pseudo-native structures of proteins, whose masses were consistent with the ones independently obtained through MALS detection [50], as well as quantitative analysis of phospholipids in plasma lipoproteins [50].

#### 1.3.5. Novel Couplings

Novel and still widely unexplored online coupling with FFF instrumentation include capillary cells (LWCCs), optical trap-based Raman flow cells, γ-ray detectors, and nanoparticle tracking analyses (NTAs). LWCC is a spectrophotometric detector characterized by a flow cell with a longer optical path length compared to traditional UV/Vis instrumentation. Its application to the analysis of AgNPs in groundwater allowed a LOD improvement of roughly two orders of magnitude, highlighting the performance of the detector in analyzing colloidal traces in environmental matrixes [51]. The development of optical trap flow cells for Raman microspectroscopy and coupling to AF4 and CF3 allowed the identification of latex and polymethylmethacrylate particles based on chemical composition, opening new possibilities in the analysis of nanoplastics in food and environmental matrixes [52]. The coupling on an γ-ray detector improved our capability of studying the interaction between radioactive components and the samples of biological/biomedical interest, such as liposomes, as well as their evolution over time [53]. Finally, the online coupling of NTA with AF4 [54] allowed us to obtain the particle count information in a simple in a fast way without suffering from the intrinsic problems of batch mode analysis (i.e., sample polydispersity, instrument sensitivity, and operator error). Within this context, particle counting with AF4-MALS is also gaining attention, although little work has been performed in this area so far [55,56].

The principal detectors hyphenated to FFF platforms are summarized in Table 3.

### 1.4. FFF and Offline Characterization

The online coupling of detectors to the FFF instrumentation represents the fastest and simplest way to work with FFF. However, to obtain a more in-depth characterization of complex samples (such as blood plasma and cell culture media), the use of offline techniques, in addition to the FFF separation/online characterization, is often required. Working with a coupled offline FFF platform requires the collection of the separated fraction which may undergo a series of pretreatments (such as concentration) before offline analysis. Figure 4 schematizes the operational phases of these platforms and reports the most common techniques coupled offline to FFF.

The reasons for offline coupling are numerous: some techniques (e.g., SDS-Page and western blotting) intrinsically cannot be coupled online with FFF, others may require a pretreatment of the fractions (e.g., mUHPLC-ESI-MS/MS), or the online version of the detector is not available in the lab (e.g., DLS). Studies exploiting offline coupling are described in-depth in the following sections.

An additional FFF separation of the collected fractions can be seen as a particular kind of offline coupling, which is sometimes exploited to improve the separation of complex samples, such as cell homogenates [57]. However, as mentioned before, this is a labor-intensive procedure since a reconcentration step of the collected fractions before the second (reinjection) run is most of the time required. HF5 is ideally suited for such operation because its (1) small channel volume and low operation flow rates allow reducing dilution and volume of the collected fractions, and (2) the relaxation/focusing step that takes place between the first and second run (refocusing) allows reestablishing the volume and concentration of the sample plug before the second elution. Within this context, Zattoni et al. further improved the workflow by developing a system that automatically reinjects the separated analytes online without requiring offline sample collection [58].

### 1.5. Interaction Studies and Immunoassay Development

FFF features such as gentle separation, the absence of a stationary phase, and versatility both in terms of mobile phase and sample injectable greatly enlarge FFF applications, in addition to the mere separation and characterization of analytes. The use of FFF platforms to study the interactions between different sample components, even in a complex matrix or in the presence of interferent agents, is, nowadays, well-established [59]. Within this context, the subsequent step is the exploitation of FFF’s ability to study component interactions and develop immunoassays. Pioneering studies exploiting an AF4 platform, micrometer-sized beads coated with the capture antibody (acting as a solid phase), and an analyte conjugate (used as a tracer) have been reported [60,61,62]. GrFFF has also been implemented into enzyme immunoassays for the detection of intact pathogenic bacteria in biological samples [63]. Recent advances in these two topics are reported in the following sections based on the nature of the samples involved.

## 2. Applications

### 2.1. Protein Analysis

ElFFF was the first FFF technique exploited to separate and analyze proteins [64]. Compared to protein electrophoresis, ElFFF stands out due to its absence of adverse heating, support effects, and higher separative power. Since then, AF4 and HF5 established themselves as the leading FFF techniques in protein analysis. The combination of DLS and MALS can greatly improve the overall characterization of the samples. The Rg/Rh ratio and the slope of the conformation plot (log(Rg)/log(Mw)) provide an indication of the shape and density of the analytes, which can improve the knowledge of the protein structure–function relationship [65,66,67].

Online coupling with MS techniques, such as inductively coupled plasma mass spectrometry (ICP-MS) and electrospray ionization/time-of-flight mass spectrometry (ESI/TOFMS), allowed for the targeted analysis of metalloproteins involved in diseases [68], the monitoring of metal NP–protein interactions [69], and MS detection of proteins in their native structure [70,71,72].

Phenomena such as protein denaturation, dissociation into smaller units, and aggregation affecting size, charge, density, shape, and biological activity can be easily observed [73]. The study of protein interactions, stoichiometry, and aggregation is thus of utmost importance. Since protein self-aggregation or non-covalent interaction with other systems are usually promoted by weak forces [59], it is difficult to preserve the relative transient complexes with the classical analytical approaches, such as SEC [74,75,76]. The gentle separation provided by FFF, and in particular AF4 and HF5, allowed to successfully quantify protein aggregates [77,78], study weak binding interaction with receptors and ligands [79,80], examine labile dynamic equilibria between various oligomeric protein forms [81], explore the stability of protein aggregates [82], and explain the effects of mutations on aggregation [83].

Protein aggregation represents the most common and troubling outcome of protein instability in all the development phases of pharmaceutical proteins and peptides [84]. A common concern related to aggregation studies in AF4 is the possible impact of the focusing step on the aggregation/oligomerization state of the analytes, which may also lead to interaction with the membrane, as it concentrates the sample close to the accumulation wall. Although the possible impact of the focusing step should always be considered, recent studies have reconsidered such an effect, downsizing its importance when applying AF4 to the separation of labile protein complexes and transient aggregate populations [81,82]. The choice of an appropriate mobile phase also plays an important role regarding protein aggregation. A poorly chosen composition of the carrier liquid can induce analyte–analyte and analyte–membrane interactions [85,86]. Consequently, pH, ionic strength, salts, and additives (e.g., metal ions, surfactants) must be carefully selected. The choice is usually made by trying to mimic the composition of the original sample matrix.

Nowadays, regulatory bodies, like FDA and EMA, require results from other techniques, like Sedimentation Velocity Analytical Ultracentrifugation (SV-AUC) and AF4, to complement SEC data. To be marketed as “alternative drugs,” newly developed formulations must exhibit the same properties as the reference drug. The type of studies that focus on assessing the existence of these relationships are called sameness studies. Within this framework, many aggregation studies of proteins and mAbs of medical/pharmaceutical interest by exploiting and comparing AF4 and SEC results have thus been published [87,88].

A comprehensive example involves liraglutide, an Active Pharmaceutical Ingredient (API) peptide used in the treatment of Type 2 diabetes and chronic obesity, with the brand names Victoza and Saxenda. A first study, exploiting an AF4-UV-MALS platform, published by Frederiksen et al., measured the aggregation state of liraglutide as hexameric (Figure 5) and compared the result with the ones stemming from theoretical calculations [89]. These evaluations were, however, performed using a mobile phase different from the formulation environment, which could greatly affect the aggregation.

Recently, a sameness study on liraglutide, taking into account limitations, has been published [2].

The developed method was able to identify the association state of liraglutide in native conditions, which was pentameric. The results shown in Figure 5 confirmed the identity between commercial Reference Listed Drug (RLD) samples, finished dose form (FDF), and APIs. The results obtained by an SEC analysis on the same samples showed a lower aggregation rate compared to AF4, which suggests the non-covalent nature of the liraglutide aggregates.

AF4 was also used to evaluate protein aggregation kinetically and monitor the effect of centrifugation, the presence of additives, and temperature [90,91,92,93]. The study of protein interactions by FFF went even further in the characterization of multi-component protein structures of greater physiological relevance than the typical one-/two-component systems [94,95].

The separation, characterization, and monitoring of protein interactions performed by FlFFF in extremely complex matrices with minimal (on none) sample preparation represent the new frontier in protein analysis. Some examples are milk [96], wine [16], yolk [97], and whole serum [45]. The latest effort in this field is represented by the study of the heme–protein interactions in the whole serum of healthy donors and IgG spiked serum, simulating pathological conditions through an HF5 multidetection platform. (Figure 6). The main interacting proteins were found to be human serum albumin (HSA), Haptoglobin (Hd), and Hemopexin (Hx), while a low amount of heme was also bound to other proteins, such as IgA and IgM. This approach also represents a proof of concept for the further investigation of selective interactions with small molecules and probes of clinical/pharmaceutical interest [98].

Recently, a completely different approach to the FlFFF of complex samples has been proposed, based on the application of principal component analysis (PCA) and other chemometric tools [99], on the fractograms of different samples of a certain food matrix [100,101]. A recent study highlighted the role of proteins in discriminating the regional provenience of tomatoes [102].

Classical tools for the analysis of proteins in complex matrixes also include the offline coupling of AF4 with other techniques, generating multidimensional platforms. SEM/TEM, western blotting, 2D-PAGE, and nLC-ESI-MS/MS are widely explored for the proteomics of extracellular vesicles (see Section 2.4).

Although AF4 and HF5 have cemented their leading role in protein analysis, in recent years, other innovative FFF platforms are emerging. For example, in 2015, Johann et al. developed ElAsFlFFF, a separative platform that combines ElFFF and AF4 [20]. This system allows separation based on electrophoretic mobility, overcoming some of the restrictions of ElFFF such as the electrode interference, the requirement of low ionic strength fluid as carrier, and the reduction in gaseous electrolysis products, thanks to the application of a low voltage.

Finally, Kenta et al. [103] reported the use of sedimentation FFF (SdFFF) to study the aggregation behavior of protein emulsions.

### 2.2. Nucleic Acids Analysis

Technologies based on nucleic acids (DNA, RNA, aptamers) have been used in biomedicine for gene therapy, drug development, vaccination, and as novel tools for the development of miniaturized devices with high affinity and selectivity to a target analyte [104].

The traditional techniques employed to study DNA and RNA strands, including DLS, analytical ultracentrifugation (AUC), TEM, SEM, and SEC, suffer from a series of intrinsic problems. DLS has limits in the evaluation of the molar mass/size distributions of highly polydisperse samples, while AUC is not a self-sufficient technique since it requires information on the specific volume of the analytes [105]. TEM and SEM imaging are expensive and slow techniques that also work on the sample after a drying process, which could alter their properties. Compared to SEC, the use of FFF techniques improved the overall characterization of RNA and DNA strands, providing higher sample capacity, shorter analysis time, and milder separation, and were able to preserve conformations and interactions with other components [19]. Moreover, the flexibility in the mobile phase composition allowed producing a stabilizing environment for the species of interest [106,107].

Although SdFFF has proven sufficiently gentle to fractionate DNA and small supercoiled plasmids without altering their conformations [108], FlFFF (AF4, HF5) was established as the leading technique in this field. AF4 was applied to the fractionation of plasmid fragments [109] and was used to elucidate the conformation of linear and circular DNA [110].

The use of cells with genetically modified metabolisms to produce industrial and pharmaceutical proteins has resulted in an increased importance of industrial cultivation [111]. The activity of ribosomes is an important parameter to monitor the growth of the engineered cells of interest. A series of studies monitoring ribosome activity through quantification of tRNA by AF4 has been published [112,113]. AF4 was also applied to the characterization of DNA systems for gene therapy. Cationic lipid DNA complexes are being studied to perform gene delivery and gene silence [114]. However, their transfection and silencing efficiencies remain low compared to engineered viral vectors. This is probably due to their heterogeneity in terms of size and net charge. Lee et al. developed the first method to separate and characterize an array of those vectors [115]. Recently, in the context of tracing nanopollutants, such as antibiotic resistance genes (ARGs), the DNA–particulate matter interactions have been studied with an AF4-ICP-MS platform [116].

Aptamers are short ssDNA or ssRNA sequences synthetized from nucleic acids, which can fold into several three-dimensional structures as a function of their sequence and binding conditions. They are increasingly studied due to their outstanding applications in the development of biorecognition-selective sensors [117]. Numerous techniques for the evaluation of aptamer–target binding are reported in the literature; the most widespread is surface plasmon resonance (SPR), SEC, and capillary electrophoresis (CE) [118,119]. Separation in SEC and CE allows for studying aptamer–target interactions with no need for immobilization, which can modify the nature and extent of the interaction [120]. An SEC packed column could, however, impact the weak aptamer–target interactions, inducing artifacts. On the other hand, the CE open channel is affected by Joule heating when current passes through highly concentrated electrolyte solutions, thus affecting electrophoretic mobility and separation [106,121]. In addition, CE has a low sample capacity. In the case of proteins, AF4 improved the overall separation and characterization of aptamers. AF4-FLD was used to isolate an aptamer–protein complex and monitor the binding between the fluorescently labeled aptamer and the target protein [107,117]. Ashby et al. estimated from the fractogram the dissociation constant (K_d_) values between the aptamer and the target; however, the values found were not consistent with the literature data (obtained by SPR) [106]. Marassi et al. reported the first method to monitor selective aptamer–lysozyme interaction exploiting an AF4-UV platform [122]. The results (Figure 7) showed that the use of specific absorption wavelengths can selectively detect the interaction between lysozyme and the intended aptamer, allowing for an efficient screening of the best aptamer for the lysozyme of interest (egg white lysozyme). The method was also suitable to understand the effect of non-specific interactions that may occur when the two components are incubated in the presence of other possible binding candidates (e.g., BSA).

### 2.3. Polysaccharides

Characterization of biopolymers is a prime challenge in biotechnology. Together with proteins and nucleic acids, polysaccharides represent one of the main families of biopolymers. The analytical information required for their characterization includes molar mass, size, and branching grade, which strongly affects polymer properties [123]. Although other FFF platforms, such as ThFFF [9], have been used, the ability of AF4-MALS to provide uncorrelated absolute mass/size characterization [124] makes it the most suited technique for polysaccharide characterization. Moreover, the hydrodynamic radius, gyration radius, and molar mass values provided by AF4-MALS also allow for branching grade determination [125].

Since polysaccharides do not usually exhibit UV/Vis absorption at characteristic wavelengths, the most common concentration detector used for their analysis is dRI. AF4-MALS-dRI has been used to characterize a wide array of polysaccharides, such as pectin [126], β-glucan [127], β-cyclodextrin [128], κ-carrageenan [129], chitosan [130], hyaluronic acid [131], and starch [132]. Further examples can be found in an extended review of the topic [133,134]. Aside from typical size/shape/PDI characterization, AF4-MALS-dRI has been also exploited to unveil the relationship between the conformation of starch and its antidiabetic activity [135] and study starch retrodegradation behavior [136,137]. Gołębiowski analyzed the changes in pectin fraction after Cu and Cd-induced gelling with the AF4-ICP-MS platform [138], while AF4 coupled with QELS was used for the characterization of hyperbranched glycopolymers produced by enzymes [139].

### 2.4. Lipoproteins and Liposomes

#### 2.4.1. Lipoproteins

Lipoproteins are complexes of lipids and proteins responsible for transporting lipids through the blood stream. They are composed of neutral lipids, such as phospholipids and proteins, on the surface. The role of lipoproteins in cardiovascular disease (CVD) has been well-studied, and their size characterization emerged as a powerful tool to assess cardiovascular health [140].

Ultracentrifugation (UC), the most common method to isolate high-density lipoproteins (HDLs) and low-/very-low-density lipoproteins (LDLs/VLDLs), is tedious, requires large sample volume, results in sample loss, and does not readily provide information on particle size. Electrophoresis [141] and SEC [142] have been used to separate lipoprotein subclasses. However, interactions with the stationary phase and restrictions on the buffer types may alter the samples [143]. The ability of FFF to separate lipoproteins was first demonstrated by Giddins [144]. Nowadays, the research is focusing on the use of AF4 [145] and HF5 [17] as a tool to isolate and analyze by powerful offline techniques, such as nUHPLC-ESI-MS/MS and different classes of lipoproteins. This provides a lipidic composition for each fraction, which can often be correlated to healthy/pathological conditions. With such platforms, it was possible to find correlations between oxidized phospholipidic profiles of LDLs and cardiovascular diseases [146,147]. Recently, an AF4 platform coupled offline with nUHPLC-ESI-MS/MS was used to identify specific lipid alterations in lipoproteins isolated from the plasma of patients with Alzheimer’s disease (AD) and Mild Cognitive Impairment (MCI) [148]. Semipreparative AF4 allowed the isolation of HDLs and LDLs/VLDLs (Figure 8). These fractions were analyzed with nUHPLC-ESI-MS/MS and showed that the total level of most lipid classes increased more than twofold in the LDL/VLDL fraction of AD patients, while the levels of diacylglycerol (DG) and phosphatidyl glycerol (PG) decreased in the HDL fraction. A statistical analysis of the result allowed the identification of a series of lipids, whose increasing abundance is correlated to an increase in brain damage level. Although extremely powerful, these methods are extremely complex, slow, and expensive. Simpler approaches are, therefore, welcome for routine clinical analysis practice. In this context, Rambaldi et al. developed an interesting, simple, and fast AF4 system combined with an online post-fractionation enzymatic detection for the single-run determination of cholesterol and triglycerides associated with each fractionated lipoprotein class [149].

#### 2.4.2. Liposomes

Liposomes are artificial small self-assembled vesicles based on phospholipids with outstanding features as drug carriers. Compared with traditional drug delivery systems, liposomes exhibit better properties such as site-targeting, sustained or controlled release, the protection of drugs from degradation and clearance, superior therapeutic effects, and lower toxic side effects [150]. Conventional liposomes consist of a lipid bilayer with an aqueous center. Further optimization to improve their drug loading, permeability, and target selectivity can be achieved by surface modifications [151]. AF4 has been exploited to study liposome stability, size distribution, shape, and drug loading/release properties, as well as monitor the ability of synthesis to control the size of the NPs. Overall multidetection AF4 emerged as a powerful asset in the early preclinical R&D development settings, as well as a reliable quality control technique for the later stages of the product development and manufacturing of liposomal-based drugs. Within this context, a robust Standard Operating Procedure (SOP) document that focused on the measurement of the physical–chemical properties of nanopharmaceuticals by multidetector AF4 (MD-AF4) has been developed [152].

Stability-wise, size-dependent liposome shrinking/swelling was observed based on its exposure to hyper- or hypo-osmotic media while being only slightly (or not at all) affected by changes in ionic strength [153]. Stability over time was studied for drug-loaded [154] and surface-modified [155] liposomes, while T.J. Evjen et al. observed the effect of ultrasound on the size/shape and drug-loading capability of two kinds of liposomes [156].

The measurement of Rg (MALS) and Rh (DLS) of the separated liposomes and the subsequent evaluation of the shape factor improved the overall characterization and monitoring of the drug-loading mechanism [157,158,159,160]. Caputo et al. calculated a range of 0.7−0.9 for the Rg/Rh ratios of PEGylated lipid micelles, which indicated a compact homogeneous core shell spherical structure, while the conformation factor emerged as being equal to 1 for empty unilamellar liposomes or lower when the encapsulated drug occupied a significant mass fraction of the aqueous core [161].

The relationship between the size and formulation of the fractionated liposomes has also been studied by applying LC-MS [162] or HPLC-UV-charged aerosol detection (CAD) [163] to the fractions separated by AF4.

Synthesis monitoring studies were performed by Lee [164] to evaluate the self-assembly of cationic lipid–DNA gene carrier complexes and by Jahnet [164] to prove the size-tune capability of a synthesis performed in a purpose-made microfluidic channel.

### 2.5. Subcellular Structures: Organelles and Exosomes

Isolation of subcellular compartments for analysis of their content is widely used in bioanalytical chemistry to develop biomedical assays [165], understand their functioning [166], and evaluate the health state of the organism of origin [167]. In the past, numerous FFF variants (SdFFF, ElFFF, FlFFF) were successfully applied to the gentle and native separation of these species. Recent trends focus on comprehensive FFF platforms to map their composition in terms of proteins, lipids, and other components of biological relevance.

#### 2.5.1. Organelles

SdFFF was the first FFF technique to be applied to the fractionation of mitochondria, microsomes, Golgi membranes, and plasma membranes, mostly preserving their structure [168]. A microfabricated ElFFF was used for separating mitochondria from whole cells and nuclei and for the separation of mitochondrial subpopulations [169]. AF4 was applied to the fractionation and size characterization of bacterial ribosome subunits. [170]. This technique is also helpful in obtaining enriched aliquots of a certain organelle through AF4 and fraction collection. Kang et al. reported the first FFF study to enrich mitochondria from rat liver with sufficient yield for additional analyses [71].

#### 2.5.2. Exosomes

Extracellular vesicles are important in cell signaling and consist of membrane-bound vesicles that have been released from cells. Compared to other extracellular vesicles, such as microvesicles and apoptotic bodies (raging between 100 and 5000 nm), exosomes are much smaller (15–50 nm) [171], too small for typical light microscopy or direct flow cytometry [172]. Their isolation is typically performed by UC [173], SEC [172], polyethylene glycol [174], density gradient centrifugation [175], and immunoaffinity methods [176]. However, UC requires considerable amounts of samples, while its efficiency in terms of time and purity is relatively low; density gradient centrifugation and immunoaffinity methods are also characterized by low yields. SEC does not prevent the interaction of the exosomes with the stationary phase while PEG, although a promising strategy, is still widely unexplored in terms of its applicability to different matrixes [177].

Due to their outstanding features (see Section 1.2), AF4/HF5 devices have also been exploited for exosome characterization. In addition to size separation, when these systems are coupled online with multiple detectors, additional information is also provided by spectroscopy and laser scattering. However, due to the complexity of the samples, a mere online coupling is not enough to distinguish different exosomes by composition nor identify the biomarkers of pathological conditions. Consequently, most of the studies exploit additional offline techniques, as described in Section 1.4. One of the main hindrances to offline coupling is the dilution of separated analytes at the AF4 outlet. Prior to further analysis, fractions often need pretreatment steps, such as extraction and concentration. An improvement is represented by the recent use of HF5 [40]. Due to its miniaturized channel, HF5 provides minimal dilution of the fractionated vesicles and requires extremely low injection volumes and is thus well-suited for the analysis of scarcely available samples. The most outstanding application in this field was reported by Zhang et al. [178]. Using an F4 platform, they identified two exosome subpopulations (large exosome vesicles, Exo-L, 90–120 nm; small exosome vesicles, Exo-S, 60–80 nm) and discovered a third abundant population of non-membranous nanoparticles termed “exomeres” (~35 nm). They also demonstrated that each of the Exo-L and Exo-S exosomes had unique N-glycosylation, protein, lipid, DNA and RNA profiles, and biophysical properties. A summary of the most successful and recent application of FFF to exosome isolation is given in Table 4.

### 2.6. Viruses and VLPs

Virus-like particles (VLPs) are multimeric nanostructures composed of one or more virus proteins in the absence of genetic material. Having similar morphology to natural viruses but lacking any pathogenicity or infectivity, VLPs have gradually become a safe substitute for inactivated or attenuated vaccines. They are also used in gene therapy and drug delivery since they are characterized by high cell penetration power [183,184]. Their use is subordinated to the quality control and purification of VLPs and the viruses from which they derive [19,185].

At present, the most widespread techniques for virus purification are UC [186], anion exchange chromatography [187], and SDS-PAGE [188]. However, UC results in low sample recoveries, while anion exchange chromatography and gel electrophoresis can be harmful to the viruses. Consequently, gentler approaches, such as FFF, have been investigated.

SdFFF was first used to separate and characterize the mass and size of virus particles [189,190] and related aggregates [191] with little to no alteration of their infectivity [192]. FlFFF was also used in the early stages of FFF development to separate and characterize viruses [39,109,193].

Nowadays, AF4 is the most common FFF variant used in virus and VLP purification and characterization. Coupling with MALS, DLS, and concentration detectors, such as UV/VIS and dRI, allowed us to study the size distribution, shape, aggregation stoichiometry, self-assembly/disassembly processes, and stability of viruses [194,195] and VLPs [196,197,198]. Since gene therapy represents one of the main application areas of these systems, AF4 was also exploited to monitor DNA encapsulation in VLPs [198,199].

Recently, Cyclical Electrical FFF (CyElFFF), an FFF subtechnique that allows for simultaneous size and superficial charge characterization, has been applied to VLPs [200]. This technique allowed for the better separation of different VLP subpopulations, differing in surface conjugated peptide, but the resolution between different aggregates of the same VLP typology turned out worse than in AF4, limiting the overall applicability of the technique.

### 2.7. Cells and Bacteria

#### 2.7.1. Cells

The application of new cell-based technologies to the diagnosis and therapy of human diseases is impacting most fields of biomedicine, including hematology, microbiology, biotechnology, molecular biology, neurology, and cancer research. Protocols span from autologous bone marrow transplantation for the treatment of advanced cancers [201] to blood cell differential analysis, which is a common procedure in clinical and research laboratories to detect infections, anemia, or leukemia [202]. Cell characterization, separation, and purification are mandatory steps for the development of diagnostic and therapeutic protocols. Nowadays, the most common cell separation techniques are centrifugation [203], electrophoresis [204], and label-based cell sorting [205,206,207], which, respectively, rely on differences in cell density, charge, or surface antigen expression. The development of label-free, gentle cell sorting techniques is important to reduce the operational costs and the impact of sorting on cell cultures. To this end, FFF platforms have been applied to tagless cell sorting. Recent applications mainly deal with the isolation of homogeneous cell populations to understand many biological processes and are used in clinical and therapeutic field. Among FFF variants, GrFFF and SdFFF are the most widely used. Some applications regarding the use of HF5, ThFFF, and DEP-FFF are also reported.

SdFFF was initially used to fractionate yeast [208], animal and human cells [209], and discriminate apoptotic and autophagic cells [210]. The use of biocompatible materials and sterilization processes allowed the preservation of cell viability after fractionation and collection [211,212,213,214]. This feature allowed downstream studies on the separated fractions to monitor tumor development, apoptosis, proliferation, vascularization, protein expression, and kinetics of the chemically mediated neurodegeneration process [215,216,217,218,219]. SdFFF was further applied to the study of cellular phenomena, such as apoptosis induction and the differentiation process, with perspectives regarding the screening of specific molecules and the development of cellular models [220]. SdFFF was also proposed as a routine method to prepare differentiated cells to better understand the differentiation process. Megakaryocytic-differentiated cells were sorted from a human erythroleukemia cell line after induction with diosgenin [221,222]. Cardot et al. proved the technological applicability of field-flow fractionation by exploiting a centrifugal field in a patent on a device able to separate live human cells from biological fluids [223].

Even the simplest FFF variant, GrFFF, shows effectiveness as a sorting technique. Due to the low investments for personnel training, operating and relative instruments, and methods maintenance, together with the potentialities displayed, the GrFFF results are particularly suited for the development of cell sorting methods by their simple integration into available cell sorting procedures and laboratories. GrFFF was first applied to fractionate populations of human red blood cells (HRBCs) [224,225]. Studies were also performed on yeast cells to evaluate physical differences between populations [226,227,228], the determination of cell viability, and commercial yeast strain characterization [229]. GrFFF was widely employed for clinical and biomedical applications. A method based on GrFFF was successfully developed to give an enriched sample of endothelial cells from the umbilical cord vein for the extraction of genetic material for further studies on specific pathologies [230]. A neoplastic cell purging from a heterogeneous mixture of human living lymphocytes was also reported; a high depletion from neoplastic cells (>98%) was found in a specific fraction [231].

In the field of cell analysis, stem cells represent a matrix of highly biotechnological interest due to their outstanding applications in regenerative medicine and personalized therapy [232]. GrFFF and SdFFF were also applied to the separation of stem cells. GrFFF was applied to the enrichment of mouse hemopoietic stem cells from bone marrow [233] and human hemopoietic stem cells from apheresis samples [234]. SdFFF shows wider applications, ranging from stem cells for cell model systems and cancer stem cells [234,235,236]. Recently, SdFFF was applied to the study of astrocyte progenitors and human-induced pluripotent stem cells (hiPSc) for the preparation of increasingly complex models for the development of new therapeutic tools in neural and brain pathologies [236,237,238,239]. Promising results were also presented for the study of cancer stem cells for the development of personalized therapies [218,240,241,242]. SdFFF coupling with an ultra-high-frequency range dielectrophoresis fluidic biosensor was demonstrated to be suitable for cancer stem cell isolation from glioblastoma brain tumors. Cancer stem cells were separated based on orthogonal properties (size, density, shape, rigidity for FFF, and dielectrophoretic properties) with interesting perspectives for the diagnosis of complex tumor populations from the patient (Figure 9) [242,243].

However, all the presented methods show limits related to low recovery and productivity, in particular for adherent stem cells, such as mesenchymal stem cells (MSCs), due to the adhesion of cells to the capillary channel during fractionation. MSCs represent a fundamental tool for regenerative and therapeutic applications. MSCs can be isolated from a variety of tissues, such as umbilical cords, endometrial polyps, menses blood, bone marrow, and adipose tissues, and they can self-renew and exhibit multilineage differentiation. Among FFF, the NEEGA-DF (Non-Equilibrium, Earth Gravity Assisted Dynamic Fractionation; patent number US8263359B2) method, for which interaction with the fractionation device is avoided by in-flow injection by the absence of stop-flow cell sedimentation and using elution flow rate values that are able to generate hydrodynamic forces that are intense enough to lift and keep cells away from the channel wall, has shown interesting results in terms of its application to stem cell sorting and adherent cells, such as MSCs. The ability of this method to identify and sort different types of cells, from mesenchymal to epithelial cells, hematopoietic cells, and different sources MSCs, was demonstrated, together with its ability to underlined intra-differences in the cell population that other techniques do not do, such as differences among stem cells isolated from different sources or within a cellular population [244]. A prototype version of a fractionation device was already demonstrated and was able to enrich and sort multipotent mesenchymal stem cells from clinical samples using the NEEGA-DF principle [245]. Recently, we developed a new automated technique, Celector^®^, which can also implement NEEGA-DF technology. Celector^®^ can taglessly characterize and sort cells, ensuring high productivity and cell recovery, which are basic requirements for cell analysis. The output of the instrument is a multiparametric fractogram representing the number, size, and shape of the eluted cells as a function of fractionation time, and it is the fingerprint of the cell sample. Celector^®^ was successfully used for the enrichment and quality control of stem cells for therapeutic applications [246,247,248,249,250,251]. In Figure 10, the schematic use of Celector^®^ is described (a). An example of its applications to the quality control of the isolation protocol of amniotic epithelial cells for regenerative use is reported (b).

Although SdFFF and GrFFF have been dominant in cell sorting, new variants, such as HF5 [252,253] and Dielectrophoresis FFF [27], have been proposed. The HF5 miniaturized and cheap channel allows for working in disposable mode on small sample amounts, avoiding carryover effects, and minimizing dilution. DEP-FFF also attracted a lot of interest in recent years due to its excellent ability to separate different kinds of cells [254,255,256]. In the literature, DEP-FFF has been exploited to separate and characterize human leukemia (HL-60) cells from peripheral blood mononuclear cells (PBMCs) [257] and cancer cells from blood cells [258,259,260]. This technique, however, still suffers from several technical limitations such as low throughput, difficult scalability, constraints in the mobile phase choice, and Joule’s heating [27].

#### 2.7.2. Bacteria

In 1991, GrFFF was the first conventional FFF technique exploited in bacterial analysis. Merino et al. showed its ability to isolate live and dead microfilariae from blood [261], highlighting the possibility to use GrFFF variants for the diagnostic screening of parasite infections. In the following years, however, SdFFF dominated the field due to its highest resolution power [262,263].

Other variants such as ThFFF [264], DEP–GrFFF [265], and HF5 [252,265] have also successfully been explored.

#### 2.7.3. FFF-MS Coupling for the Analysis of Bacteria and Cells

The outstanding features of MS coupling to FFF for the characterization of subcellular structures also translate to cells/bacteria through top-down proteomic approaches. The use of low-fragmentation ion sources, such as matrix-assisted laser desorption–ionization (MALDI), allows for desorption and detection of intact proteins expressed on cell surfaces, which can be used as fingerprints to identify different typologies of cells within the separated fractions. Lee reported the first use of SF4 coupled offline with a MALDI–TOF–MS detection system to analyze bacteria [266]. The FlFFF–MALDI–TOF–MS method was then improved using HF5 due to its intrinsic advantages over AF4 [267]. Moreover, a typical FFF-ICP/MS platform was used to distinguish bacteria subtypes by studying their different metal sorption, which provides orthogonal information to their size/shape characterization (Figure 11) [268].

### 2.8. Engineered and Naturally Occurring Nanoparticles (ENPs/NNPs) for Nanomedicine

ENPs and NNPs with controlled size distribution (usually below 100 nm) attract a lot of interest due to their applications in drug delivery systems and their applicability in novel therapies, such as gene and photothermic therapy [11]. Major limitations to their actual applications are due to the lack of pure and well-characterized NPs, which are partially related to the need for robust routine methods for their quality control and characterization. Offline DLS and TEM/SEM microscopy are the most common techniques used for this purpose. Their key limitations are overcome by FFF platforms due to their ability to work in native conditions, with minimal to no sample preparation and gentle separation. Due to the size range of those systems, AF4 represents the most common FFF variant successfully exploited for analysis. Since size determines the NP properties (ex., cellular penetration), DLS and MALS (as well as concentration detectors) are basically always coupled to the separation system. However, since ENPs and NNPs cannot always be discriminated by size alone, composition-sensitive detectors, such as ICP-MS and spICP-MS, are gaining importance [269,270,271].

Typically, ENPs include metallic, metal-oxide, and polymeric NPs, while NNPs range from biopolymers and dissolved organic matter to colloidal inorganic species. This section is focused on the application of multidetection FFF to study the most common ENPs and NNPs in biotechnology and nanomedicine.

#### 2.8.1. ENPs

Metallic, metal-oxide, and polymeric NPs represent a wide area of study for medical applications such as bioimaging, biosensors, targeted drug delivery, photoablation, and radiation therapy [11,272]. The typical configuration used to characterize ENPs and study their stability and behavior in complex media comprises AF4 system coupled by a MALS and at least one concentration detector (however, sometimes AF4-DLS platforms are used [273]). This kind of configuration is extremely simple, non-destructive, and versatile, and is thus easily applicable to a wide range of ENPs such as gold and silver NPs [274,275,276], metal oxide NPS (ex., ZnO) [277], MOFs [278], fullerenes [279], quantum dots [280], nanoplastics [281,282], and polydopamine nanosystems [283].

An AF4-ICP-MS is also often exploited for the characterization of those systems, although it is more expensive and destructive for this platform to offer more in-depth characterization of engineered NPs. Since this platform is able to detect even small traces (ppb) of metals, it is particularly suited for the analysis of metal-based ENPs [269,270,271,284,285], allowing us to evaluate the stoichiometry of the metal ligand (ex., peptides, immunoglobulins) complexes [45,69,286,287].

HF5 platforms have been exploited to monitor ENP synthesis [283], size [288,289,290], and stability [291], and observe their interactions with other components [292,293].

Although FlFFF (AF4 or HF5) is the mostly used FFF typology for the characterization of inorganic nanoparticles, other variants, such as ThFFF [294,295,296,297,298] and SdFFF [299], are still currently used because they provide separation according to different NP properties. ThFFF separates based on the thermal diffusion coefficient, which is influenced by the surface and bulk composition of the analytes, while SdFFF exploits effective mass differences between the analytes.

#### 2.8.2. NNPs

Organic nanomaterials show wide applications in several medical domains such as molecular imaging, pharmaceutical formulations, and image-guided therapies. Moreover, unlike most inorganic nanomaterials, these materials are characterized by easier tunability in composition and surface functionalization, excellent biocompatibility, and controlled release (ex., by self-degradation) [300]. Since particle size distribution (PSD) greatly affects the performance of these systems, FFF (especially AF4) platforms have been exploited for their evaluation. AF4 studies are thus reported on some of the most common organic nanosystems such as poly(lactide) (PLA) and poly(lactide-co-glycolide) (PLGA) NPs [301,302,303], dendrimers [304], and glycopolymers [305].

Nanogels are colloidal particles consisting of solvated, crosslinked polymeric networks and represent one of the most interesting families of organic nanomaterials [306]. They are characterized by high sensitivity to stimuli and degradability. These NPs typically undergo a deswelling transition at temperatures close to the human body, paving the way to potentially traceless drug delivery systems [307]. Moreover, nanogels were shown to provide the triggered release of macromolecules [308]. Poly(N-isopropylacrylamide) (pNIPAM), one of the most common nanogels, was first characterized with AF4 by Smith et al. [309], while Niezabitowska et al. investigated the effect of the synthetic protocol on the internal structure [310]. AF4 was also used to monitor pNIPAM degradation [309] and separate the nanogel from its degradation product, highlighting the ability of AF4 to function as a purification/quality control step before clinical application [311,312].

## 3. Conclusions

As mere separative systems, FFF platforms are characterized by high operational ranges and possess a series of features unmatched by other separation techniques:(1)Versatility in terms of both mobile phase and injectable sample that allows working under conditions as close to native conditions as possible;(2)The absence of a stationary phase prevents unwanted interactions.

These features allowed for the exploitation of FFF in the separation of extremely complex samples of biological/biotechnological interest, such as whole serum and cellular lysates, without any pretreatment. Online coupling to a series of detectors (UV/VIS, FLD, dRI. MALS, DLS, ICP-MS, ESI-MS/MS) greatly enhances the platform’s ability to evaluate the size, morphology, and chemical composition of separated analytes. They also support the study of complex phenomena such as interactions in complex matrixes, encapsulation efficiency, loading properties, and the biological activity of cells, bacteria, and organelles. FFF multidetection systems are also gradually expanding their application beyond academic research. For example, regulatory bodies, like the FDA and EMA, require comparing quality control results from multiple techniques. Hence, FFF routinely flanks SEC to provide complementary results for drug development. The fractioning of the gently separated analytes and subsequent offline analysis with other techniques allow the full expression of FFF’s analytical power. This kind of approach allowed the mapping of extracellular vesicles and the determination of the composition profiles (in terms of proteins, lipids, N-glycans, and nucleic material) of complex biological systems such as cells, vesicles, and bacteria, and the monitoring/evaluation of health conditions of the corresponding organisms.

Major concerns for the FFF applications to biological systems are related to the need for high-throughput methods that are able to process large numbers of biological samples, as well as the sterility issues requested for the further use, analysis, and profiling of biological samples.

A persistent problem is the limited injectable amount for most channels and the collection of fractions that are too diluted for offline analysis due to sample dilution in the channel. The development of semipreparative FFF channels (which allow higher injection volumes) or miniaturized systems that reduce sample dilution is an improvement in this field. The development of a dedicated, preparative FFF system, however, is still an unsolved challenge. Sterility conditions can be achieved by flushing the overall FFF system with decontamination and sterilization liquids. In some cases, it is necessary to consider placing the instrumentation under a laminar flow hood to ensure the total sterility of the processed material. Although some protocols and instrument set up adjustments were proposed, for some FFF instrumentations, the absolute control of sterility is still an open issue.

Other challenges that must be tackled to make FFF also effective for biological samples include the development of robust methods. Sample–membrane compatibility is a strict requirement to provide reproducible separation since their material and cut-off properties critically influence potential sample loss and sample recovery. The availability of different membranes can highly increase specific applications to biological samples, even when considering the broad variety of analytes present in the biological matrix. In addition, standardized protocols are required for specific high-impact applications.

Finally, comprehensive analytical approaches using FFF coupled with other techniques have shown an ability to increase the role of FFF in biotechnology. The improvement of coupling to MS detectors and the integration of mathematical tools in light-scattering detectors to gain more specific morphological characterization, even for non-spherical analytes, can further accelerate this process by providing an increase in biological applications (i.e., the study of highly complex protein structures and aggregation status, gene assays for gene based-therapies, different vesicles profiling for clinical applications, the cell shape in the differentiation process for the cell-based product in medicine).

Overall, the latest outstanding research results, the increasing number of FFF-related articles over the years (as shown in Figure 12), and the growing interest in the topic, even outside, suggest that FFF will likely play a key role in advancing numerous fields, such as molecular biology and biotechnology.

## Figures and Tables

**Figure 1 molecules-28-06201-f001:**
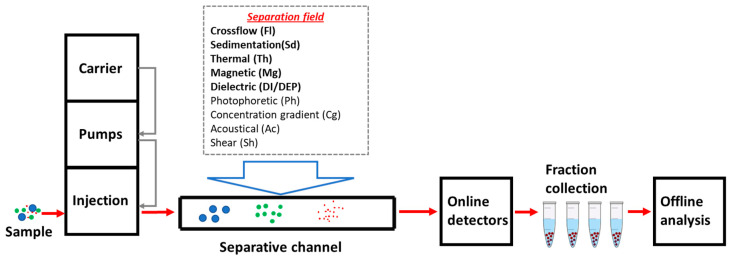
Schematic representation of an FFF platform. The separative channel is a schematization of the typical FFF channel. The dotted box reports the possible fields that have been theoretically studied to be exploited for separation in FFF. Only the ones in bold are, nowadays, significantly used.

**Figure 2 molecules-28-06201-f002:**
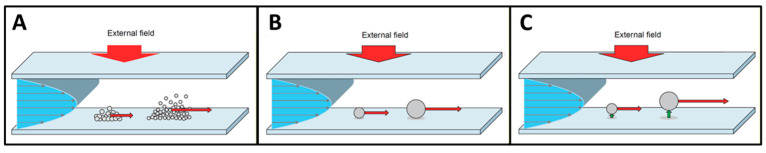
Schematization of the evolution modes in FFF. (**A**). Normal mode (**B**). Steric mode (**C**). Hyperlayer mode. Image adapted from [5].

**Figure 3 molecules-28-06201-f003:**
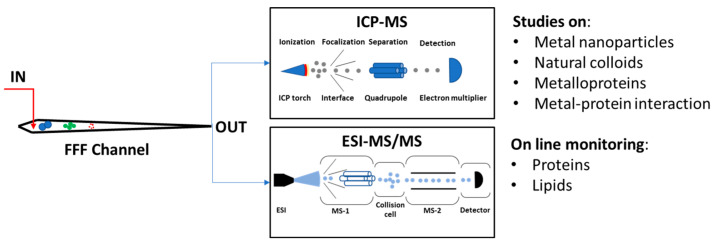
Main applications of FFF-MS online coupling.

**Figure 4 molecules-28-06201-f004:**
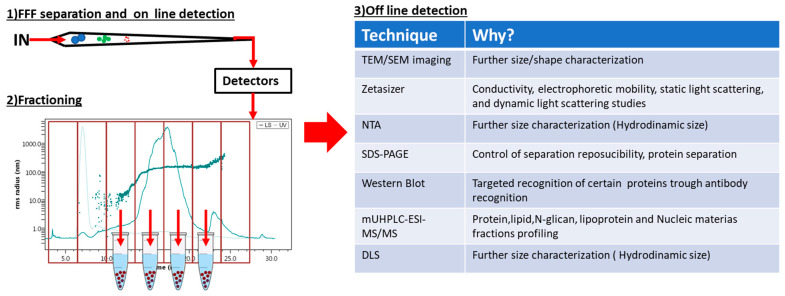
Schematization of a coupled offline FFF platform.

**Figure 5 molecules-28-06201-f005:**
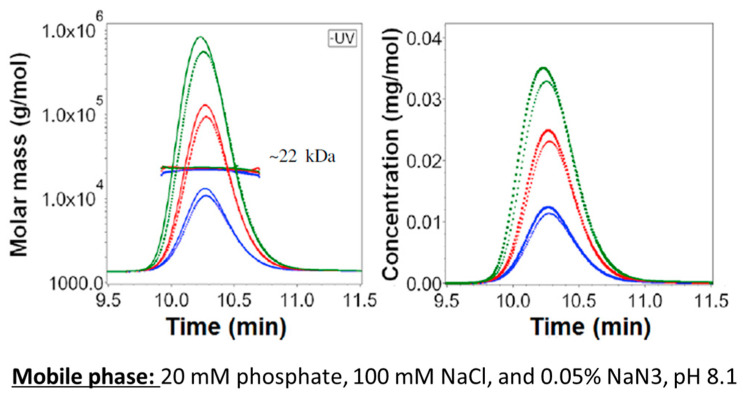
AF4-UV-MALS analysis of undiluted Victoza (6 mg/mL liraglutide) and 10-times diluted Victoza (dotted peaks). Different injection volumes were tested. All analyses showed a single peak in the UV chromatogram and a uniform molar mass of ~22 kDa across the peak. Different colors refer to different injection amounts. Image adapted from [89].

**Figure 6 molecules-28-06201-f006:**
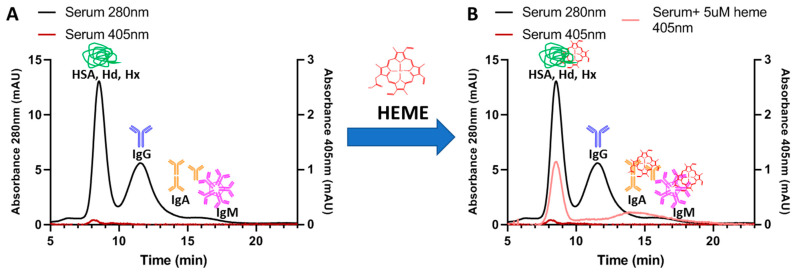
Application of an HF5-based platform as a tool for evaluating the selective interactions of components (heme) in a complex matrix (whole serum). (**A**). The fractogram shows the profiles of separated serum components at two wavelengths: 280 nm (protein specific) and 405 nm (heme-specific). (**B**). The 405 nm signal, registered after spiking serum with heme, simulating hemolysis, suggests the preferential binding of heme to HSA, Hd, and Hx. Binding to IgA and IgM has also been observed. Image adapted from [98].

**Figure 7 molecules-28-06201-f007:**
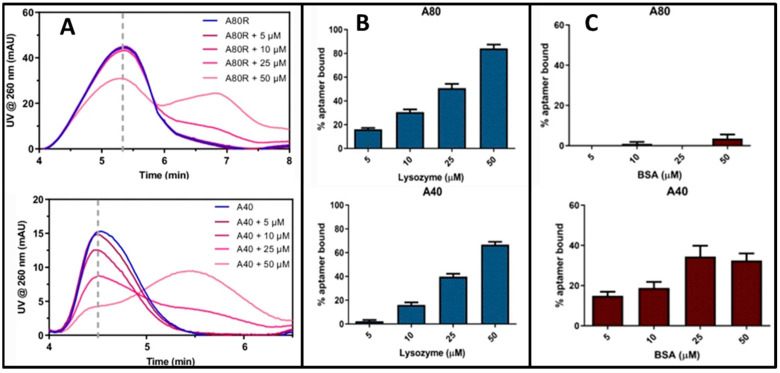
A comparative study of the selective aptamer (A80R and A40)–lysozyme (egg white) interactions exploiting an AF4-UV platform. (**A**). AF4-UV fractograms of A80R and A40 and their mixtures with lysozyme. The dashed line represents the retention time at which that signal intensity is recorded to evaluate the signal decrease in free aptamer, correlated to the formation of a complex. (**B**). Percentage of bound aptamer (expressed as % (*I_apt alone  _*− * I_apt mix_*)/*I_apt alone_*) for each aptamer at different lysozyme concentrations. (**C**). Percentage of bound aptamer to BSA (interfering agent) at different BSA concentrations. Overall, boot A80R shows better binding properties to the lysozyme, and it is less affected by the presence of BSA compared to A40. Image adapted from [122] and published with permission.

**Figure 8 molecules-28-06201-f008:**
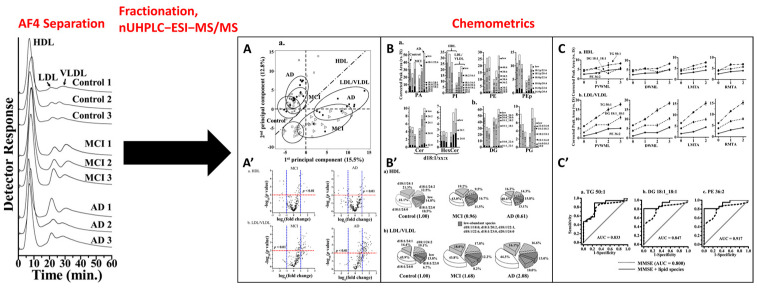
An example of the outstanding potential of offline AF4-nUHPLC to the lipidomic analysis of lipoproteins, which allowed the recognition of 363 lipidic species. PCA (**A**) and Vulcano (**A′**) analysis allowed a fast preliminary identification of lipids to differentiate the samples. (**B**,**B′**) represent absolute and percent composition of the lipids of eight different classes in the MCI, AD, and control samples. More than twofold differences in abundance were observed for certain lipidic classes when comparing the LDL/VLDL of the AD and MCI to the control ones. (**C**) Represents the correlation between the abundance of three prominent lipid species in HDLs and LDLs/VLDLs with certain diseases. (**C′**) Represents the ROC curves of the four candidate markers: two in MCI in combination with MMSE scores (solid line) superimposed with those of MMSE scores alone (dotted line) [148].

**Figure 9 molecules-28-06201-f009:**
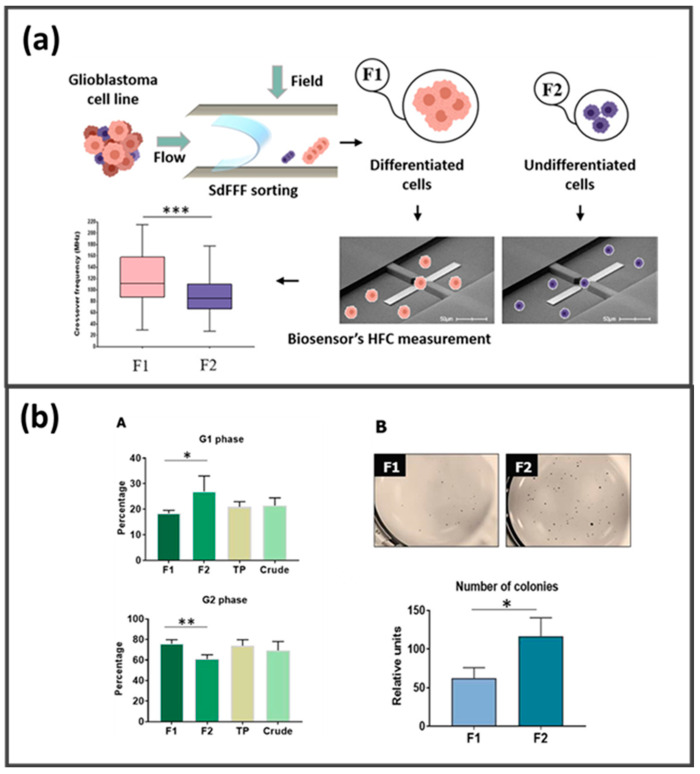
(**a**) SdFFF sorting scheme of glioblastoma cell lines: F1 = subpopulations with specific differentiation characteristics; F2 = cancer stem cells (CSCs) and high-frequency crossover (HFC) values. (**b**) Functional characterization of F1, F2, total peak collected (TP), and unfractionated cells (crude). *** *p* < 0.0001 (ANOVA). (**A**) Cell cycle analysis by DNA content measurement: CSCs (F2) are found in the G1 phase (quiescent cells). (**B**) Soft agar assay for colony formation examination: the F2 subpopulation consists of a population of cells enriched in CSCs, whereas the F1 subpopulation is enriched in differentiated cells. * *p* < 0.05; ** *p* < 0.001 (Student’s *t* test). Images adapted from [242].

**Figure 10 molecules-28-06201-f010:**
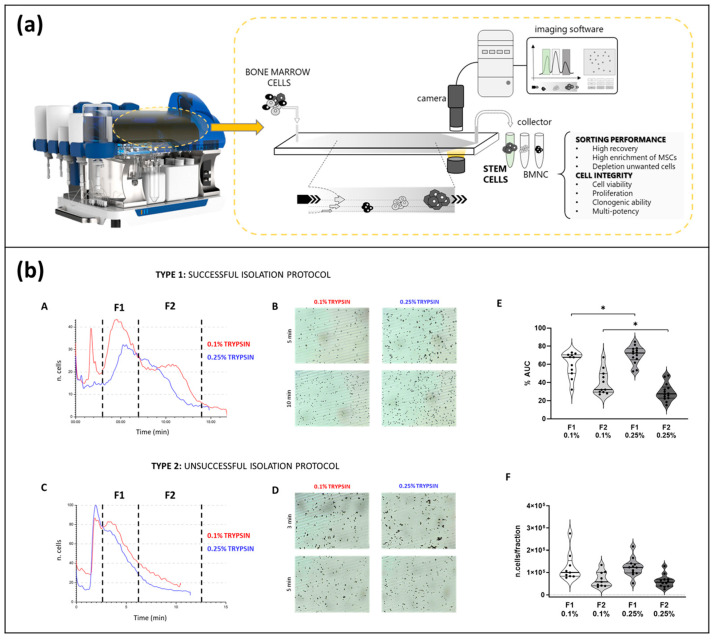
(**a**) A schematic representation of Celector^®^: cells are injected into the inlet of the separation channel filled with mobile phase and eluted through it, acquiring different velocities related to their physical properties. A camera connected to the imaging software that plots the number of counted cells vs. time (fractogram) visualizes cells. Finally, cells are collected at the outlet and divided into different tubes according to the sample’s fractogram. Images are adapted from [249] and published with permission. (**b**) Representative images of a successful isolation protocol of amniotic epithelial cells (AECs) (Type 1) and an unsuccessful (Type 2) protocol. The profile represents the number of cells versus time of analysis (**A**,**C**) and collected subpopulations F1 and F2; live images of eluting cells (**B**,**D**); cell distribution between F1 and F2 based on the calculation of the area under the curve (AUC) expressed as a percentage compared to the total area of the profile (**E**); distribution was also expressed as a number of counted cells by the software for each fraction of all samples analyzed (**F**). (*t*-test: *p* < 0.05 *.) Images are adapted from [249,250] and published with permission.

**Figure 11 molecules-28-06201-f011:**
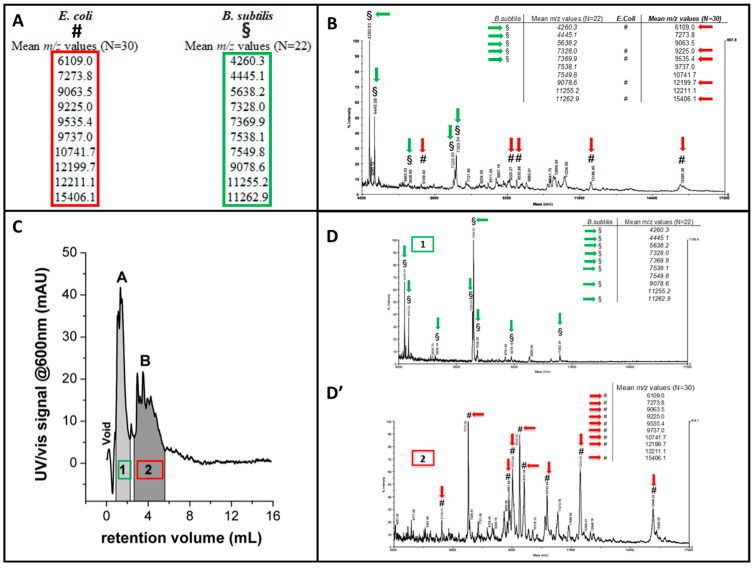
An example of an HF5 platform coupled offline with a MALDI/TOF detection system used to separate and characterize different cell subtypes. (**A**). Characteristic MALDI/TOF MS peaks of *E. coli* (§, green) and *B. subtilis* (#, red). (**B**). A MALDI/TOF MS spectrum of a mixture of the two species without HF5 separation. Five characteristic peaks were identified for boot species. (**C**). A fractogram of the same mixture separated through HF5. Two fractions were collected (namely 1 and 2). (**D**,**D′**) A MALDI/TOF MS spectrum of the collected fractions after a concentration process. The comparison with the characteristic peaks for both species allowed the identification of *Fraction 1* as *B. subtilis* and Fraction 2 as *E. coli.* A higher number of characteristic peaks was observed for both species after HF5 separation. Images are adapted from [267].

**Figure 12 molecules-28-06201-f012:**
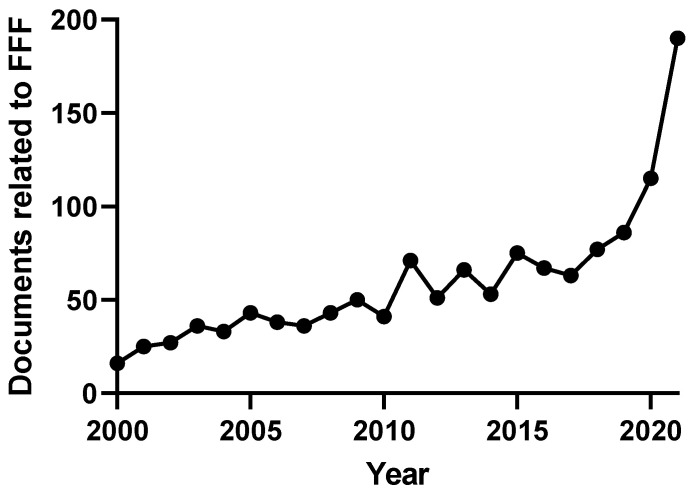
An indicative number of documents (review, research articles) published in the XXI century related to field-flow fractionation. The numbers are derived from www.sciencedirect.com (accessed on 15 September 2022) using “FFF” and “Field Flow Fractionation” as keywords.

**Table 2 molecules-28-06201-t002:** Slopes and corresponding conformation provided by the most common plots are used to evaluate particle shape in FFF experiments.

Rg/Rh	*v*-Value	Conformation
<0.7		Highly expanded macromolecule
0.775	0.33	Homogeneous-mass sphere
1.0–1.5		Branched
1.5–2.1	0.5–0.6	Random coil
>2	1	Rod-like, elongated

**Table 3 molecules-28-06201-t003:** Main features provided by the most common detectors coupled online with FFF techniques.

Detector	Features
UV-Vis/DAD	Spectroscopical Characterization Concentration detector Studies on evolving system (drug loading and releasing, conjugations…) Versatile signal (a specific)
FLD	Spectroscopical Characterization Studies on evolving system (drug loading and releasing, conjugations…) Very selective signal (quite specific) Stable signal less prone to drifts and interference
dRI	Concentration detector Unstable signal significantly affected by pressure changes Universal signal
MALS	Requires concentration signal Accurate determination of molecular weight/molar mass, radius of gyration (Rg) and fractal radius (δ) Information on the aggregation rate of nanosystems Information of the shape of nanoparticles (conformation plot) Working size range(1 nm–100 μm)
DLS	Determination of hydrodynamic radius (Rh) Estimation of the shape factor (Rg/Rh) Working size range(1 nm–100 μm)
ICP-MS	Studies on evolving system by monitoring elemental composition Low detection limit (ng/L)
High-Resolution MS	Accurate information on molar mass of molecules and complex Evaluation of loading processes inside nanosystems (e.g., Liposomal carriers)

**Table 4 molecules-28-06201-t004:** Summary of the main studies on exosomes using FFF. The table highlights the experimental set ups, including FFF platforms and additional offline techniques.

Exosome Matrix	FFF Platform	Additional Techniques (Offline)	Results	
Exosomes form human neural stem cells.	mFI-AFlFFF-UV (miniaturized frit inlet asymmetrical FlFFF UV-coupled)	TEM; LC-ESI-MS-MS	Exosome subpopulations larger than ∼50 nm were morphologically distinct from those smaller than ∼50 nm. Each exosome fraction showed a different protein pattern.	[179]
Non-labeled B16-F10 exosomes from an aggressive mouse melanoma cell culture line.	AF4-UV-MALS	DLS; TEM	Label-free separation of exosomes into subfractions and corresponding size characterization.	[171]
Lyophilized exosome standard HBM-BLCL21-30 [55] purified from the culture supernatant of an EBV-transformed. lymphoblastoid B cell line (HansaBioMed, Tallinn, Estonia).	AF4-UV-MALS	DLS; NTA (nanoparticle tracking analysis); TEM	Significant influence of crossflow conditions and channel thickness on fractionation quality. Identification, separation, and size-characterization of two exosomes subpopulations.	[55]
Exosomes isolated form the human urine of Pca patients and healthy controls.	AF4-UV	TEM; Western Blotting; UPLC-ESI-MS/MS	Exosome separation and size distribution characterization. The Lipidomic analysis of selected fractions indicated differences in lipidic content and composition between the exosomes of patients and health controls.	[167]
B16-F10 melanoma-derived exosomes.	AF4-QUELS-UV	NTA; TEM; Mobius Zetasizer AFM; Blotting and MS Techniques; Odyssey Imaging System	The separation of two discernible exosome subpopulations, Exo-S and Exo-L, and the identification of distinct exomeres, which differ in size and content from other reported particles. Proteins, glycans, lipids, and nucleic acids are selectively packaged in exomeres.	[178]
Purified human A375 melanoma exosomes.	Cy-El-FFF-UV-MALS	Mobius Zetasizer	The effect of buffer solution composition and dilution on exosome properties and separation.	[180]
Extracellular vesicles from human plasma.	AF4-UV-MALS	Western Blotting; TEM; HPLC-C18; LC-ESI-MS/MS	Human plasma contains more EVs than the paired serum and shows age- and gender-independent individual variability of the amount of EVs in human plasma. Most of the proteins identified in the EVs from human plasma were involved in extracellular matrix structural constituents and associated with the ECM–receptor interaction pathway.	[181]
Exosomes isolated from human serum samples.	AF4-UV-MALS	DLS; Western Blotting; nUHPLC-ESI-MS/M	The evaluation of the ability of ultrafiltration and ultracentrifugation in exosome isolation from serum. A simple centrifugation followed by UF offered advantages, such as faster preparation and higher exosomal recovery, with smaller sample volumes than the UC method. However, the removal of lipoproteins seemed more efficient with UC than UF.	[177]
Samples obtained from the ultracentrifugation of the culture medium of murine myoblasts (C2C12).	HF5-UV-FLD-MALS	NTA; TEM; Western Blotting	The overall characterization of small and large EVs in all the fractions obtained through ultracentrifugation. The identification of an otherwise-hidden rod-shaped species carrying nucleic content, was found predominantly in the densest SEV fractions, which could potentially correspond to exomeres.	[40]
Fractions of exosomes and microvesicles were isolated from the culture media of DU145 cells using a series of centrifugation methods, including UC.	AF4-UV-MALS	nUHPLC-ESI-MS/MS; TEM; Western Blotting	Both UC and UF methods can be utilized for the initial isolation of EVs from cell culture media prior to the FlFFF separation of exosomes and microvesicles; however, UF was found to be more efficient than UC. The hyphenation of FlFFF with ESI-MS/MS allowed for the selective detection of lipid targets and specific biomarkers.	[182]

## Data Availability

No new data were created or analyzed in this study. Data sharing is not applicable to this article.

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
