# Peer review of "Field-Flow Fractionation in Molecular Biology and Biotechnology"

_molecules, 2023, doi:10.3390/molecules28176201_

Round 1
Reviewer 1 Report
An extensive review article on field flow fractionation is presented. The topic of the paper is interesting, and the paper is well written. Some minor editing of English language and grammar are needed. I could recommend certain corrections that would, I believe, enhance the paper quality:
In paper title, abbreviation FFF should be replaced with full method name. From key words, field flow fractionation (FFF) should be excluding.
In Figure 1, the authors claim that bolded field names are used significantly. Does this mean that there are also commercially available devices? What is the device flexibility for FFF? Can different fields be used on the same device? Additional clarifications or appropriate references related to the technical details of the instruments are needed.
Some minor editing of English language and grammar are needed.
Author Response
The authors wish to thank the referee for their appreciation of the work and helpful comments, all of which were considered and accepted. The manuscript has been revised as follows:
Point 1: In paper title, abbreviation FFF should be replaced with full method name. From key words, field flow fractionation (FFF) should be excluding.
Response 1: Accepted, title and keywords are modified as suggested
Point 2: In Figure 1, the authors claim that bolded field names are used significantly. Does this mean that there are also commercially available devices? What is the device flexibility for FFF? Can different fields be used on the same device? Additional clarifications or appropriate references related to the technical details of the instruments are needed.
Response 2: Section 1.1 now provides more information on the instrumental specificity of different subtechniques, their diffusion and commercial availability. SInce a detailed description of the technical aspects related to instrumentation is beyond the scope of this review, the reader is referred to the literature for further details.
Comments on the Quality of English Language
Point 3: Some minor editing of English language and grammar are needed.
Response 3: The manuscript was carefully revised and edited to correct errors and improve the quality of language
Reviewer 2 Report
The authors review the application of filed-flow fractionation in molecular biology and biotechnology. The topic is well comprehensively reviewed. The work merits publication in Molecules after considering the following points to improve the quality of the manuscript.
The literature review period covered by this work should be defined (probably up to the end of 2022).
In the graphical abstract the application of FFF in each of the 5 mentioned fields in the circle should be at least relatively representative to its section.
For all Tables captions should show above
In Table 1 references for applications should be added
Limitation of FFF for biological sample should be discussed
The synonymous name of Sedimentation FFF (centrifugal FFF) should be mentioned
When speaking about (FFF/ICP-MS) the application for natural-colloid analysis should be better discussed.
Coupling of new detection schemes as liquid waveguide capillary cell and optical-trap-based Raman flow cell to asymmetric FLFFF should mentioned and briefly described.
The language is understandable and good enough for publication.
Author Response
The authors wish to thank the referee for their appreciation of the work and helpful comments, all of which were considered and accepted. The manuscript has been revised as follows:
Point 1: The literature review period covered by this work should be defined (probably up to the end of 2022).
Response 1: A reference to the time interval mainly covered by the review has been added in the abstract.
Point 2: In the graphical abstract the application of FFF in each of the 5 mentioned fields in the circle should be at least relatively representative to its section.
Response 2: The graphic abstract has been revised to make it more closely match the organization into sections of the applications
Point 3: For all Tables captions should show above
Response 3: Done
Point 4: In Table 1 references for applications should be added
Response 4: Done
Point 5: Limitation of FFF for biological sample should be discussed
Response 5: A discussion of the limitations and potential future improvements in the applications FFF to biological samples was added to the Conclusions.
Point 6: The synonymous name of Sedimentation FFF (centrifugal FFF) should be mentioned
Response 6: Done. See the relevant paragraph in Section 1.1.
Point 7: When speaking about (FFF/ICP-MS) the application for natural-colloid analysis should be better discussed.
Response 7: The application of FFF-ICPMS to colloids of environmental origin is now discussed in Section 1.3.4. A literature reference has been added, and Figure 3 has been modified accordingly.
Point 8: Coupling of new detection schemes as liquid waveguide capillary cell and optical-trap-based Raman flow cell to asymmetric FLFFF should mentioned and briefly described.
Response 8: New section 1.3.5 is devoted to innovative couplings between FFF and detection and characterization techniques including Raman spectroscopy, gamma ray and NTA. Appropriate literature references have been added.